

# Brief communication: Improved simulation of the present-day Greenland firn layer (1960-2016)

Stefan R. M. Ligtenberg [1], Peter Kuipers Munneke [1], Brice P. Y. Noël [1], and Michiel R. van den Broeke [1]

[1]Institute for Marine and Atmospheric research Utrecht (IMAU), Utrecht University, Utrecht, The Netherlands

*Correspondence to:* Stefan Ligtenberg (s.r.m.ligtenberg@uu.nl)

**Abstract.** By providing pore space for storage or refreezing of meltwater, the Greenland ice sheet firn layer strongly modulates runoff. Correctly representing the firn layer is therefore crucial for Greenland (surface) mass balance studies. Here, we present an improved simulation of the Greenland firn layer with the firn model IMAU-FDM forced by the latest output of the regional climate model RACMO2, version 2.3p2. In the percolation zone, much improved agreement with firn density and temperature observations is found. A full simulation of Greenland firn at high temporal (10 days) and spatial (11 km) resolution is available for the period 1960–2016.

## 1 Introduction

Since the early 1990s, the Greenland ice sheet (GrIS) has been losing mass (Van den Broeke et al., 2016; McMillan et al., 2016) and is currently one of the largest individual contributors to global sea level rise (Chen et al., 2017). During this period, the partitioning of the mass loss between decreasing surface mass balance (SMB) and increasing ice discharge has shifted from close to 50/50 between 2000 and 2005 to runoff dominating the GrIS mass loss over the last decade (Enderlin et al., 2014; Van den Broeke et al., 2016). It is likely that this trend continues in a future warming climate, making it of vital importance to model the GrIS SMB correctly.

A key process in GrIS SMB is the retention of liquid water input (surface meltwater and rainfall) that mitigates the amount of runoff by either refreezing or storing liquid water in the GrIS firn layer. Recently, some features have been discovered that enhance our understanding of meltwater retention: water is stored year-round in firn aquifers (Forster et al., 2014; Miller et al., 2017); and partly impermeable ice lenses cause lateral transport of water while the underlying firn column remains unsaturated (Machguth et al., 2016). Currently, about 45% of the liquid water input is estimated to be retained (Steger et al., 2017a, b).

With a firn densification model (IMAU-FDM), Kuipers Munneke et al. (2015) simulated the temporal evolution (1960–2014) of the GrIS firn layer. Using density observations from firn cores to evaluate the simulation, they found that model performance in the interior was good, but that the agreement deteriorated with increasing melt rates. Two possibilities for this mismatch were suggested: 1) a too simplistic representation of liquid water processes in IMAU-FDM or 2) errors in the atmospheric forcing from the regional climate model (RACMO2.3, Noël et al. (2015)). The availability of an updated atmospheric forcing (RACMO2.3p2, Noël et al. (2017)) allows us to investigate the impact of the latter. This RACMO2 update resulted in significantly more accumulation inland and less surface melt improving agreement with SMB observations in both





the accumulation and ablation zone (Noël et al., 2017). Since accumulation and surface melt are defining climate variables for the state of the firn, it is expected that the new atmospheric forcing has a marked effect on the simulated GrIS firn layer. Here, we present the new IMAU-FDM simulation and evaluate it using firn density and temperature observations.

## 2 Methods and Data

### 2.1 IMAU-FDM

A detailed description of IMAU-FDM is available in previous publications (Ligtenberg et al., 2011; Kuipers Munneke et al., 2015; Lundin et al., 2017) and will only be briefly summarised here. IMAU-FDM simulates the time evolution of firn density, temperature, liquid water content, and surface elevation in a 1-D column, forced at the surface by sub-daily (3- or 6-hourly) atmospheric output from the regional climate model RACMO2 (see below). Firn compaction is calculated using the densification equations of Arthern et al. (2010), with region-specific additions for Antarctica (Ligtenberg et al., 2011) and Greenland (Kuipers Munneke et al., 2015). Liquid water from rain or surface melt can percolate into the firn, where it is either refrozen or stored depending on firn temperature and pore space. An equilibrium initial firn column is obtained by looping over the 1960–1979 climate until the entire firn column is fully refreshed (Kuipers Munneke et al., 2015). After this spin-up, the transient simulation run starts. The IMAU-FDM simulations forced with RACMO2.3 (FDM2.3 hereafter) and RACMO2.3p2 (FDM2.3p2 hereafter) cover 1960–2014 and 1960–2016, respectively.

### 2.2 RACMO2 forcing

The atmospheric forcing of IMAU-FDM is provided by the regional climate model RACMO2 (Van Meijgaard et al., 2008), of which output of versions v2.3 and v2.3p2 are used here. Forcing consists of prescribing various SMB components (solid and liquid precipitation, surface and drifting snow sublimation, drifting snow erosion, and surface melt), surface temperature ($T_s$), and 10-m wind speed on the native 11-km RACMO2 grid. RACMO2.3p2 (Van Wessem et al., 2017; Noël et al., 2017) is the updated version of RACMO2.3 (Noël et al., 2015) and includes several changes: updated glacier outlines, topography and ice albedo fields; tuned cloud scheme parameters that increase precipitation towards the GrIS interior, correcting the underestimation of inland accumulation in RACMO2.3; modified snow properties, i.e. lower soot concentration and smaller grain size of refrozen snow, that significantly reduce melt production in the percolation zone. For the firn simulations, the most important changes are that inland precipitation on the GrIS increases by 5-10%, whereas surface melt along the margins is significantly reduced by up to 50%, leading to a higher ice-sheet integrated SMB at 11-km horizontal resolution. Statistical downscaling to 1-km resolution provides a better representation of runoff on low-elevation outlet glaciers and in narrow ablation zones. As a result, the downscaled SMB agrees better with in-situ and basin-scale SMB observations (Noël et al., 2017).





## 2.3 Firn observations

Model output from IMAU-FDM is evaluated using firn density and temperature observations from across the GrIS. Vertical profiles of firn density are compared to 62 firn cores of varying depth (8-120 m) and with locations distributed over the GrIS, although the drier northeast is slightly underrepresented. See Figure 2 in Kuipers Munneke et al. (2015) for core names and locations, which cover a wide range of melt and accumulation conditions found on the GrIS. Furthermore, deep-firn tempera-
tures (at 10 m depth, $T_{10m}$) in combination with firn density observations along a transect in western Greenland (Harper et al., 2012; Humphrey et al., 2012) are used to analyse the differences in the percolation zone in more detail. The firn air content (FAC) is used as an integrated measure for the amount of pore space present in a firn column and defined as the vertically integrated difference of the firn density and the ice density (taken to be $917\,\mathrm{kg\,m^{-3}}$).

## 3 Results

Figure 1 shows how FDM2.3p2 generally improves the simulated density profiles, compared to FDM2.3. The firn core locations can be separated into three categories based on the melt-accumulation ratio ($R_{MA}$): 1) the dry snow zone ($R_{MA} < 0.05$), 2) locations that experience moderate melt ($R_{MA}$ between 0.05-0.5), and 3) high melt locations ($R_{MA} > 0.5$). In the first and third category only small differences are noted; the biggest improvements are found in the second category.

For the dry snow zone (example in Figure 1B), the higher accumulation rates in RACMO2.3p2 result in slightly higher compaction rates and therefore denser firn in FDM2.3p2. Overall, the agreement with observed FAC in the dry snow zone is slightly worse for FDM2.3p2 ($r^2 = 0.98$ and RMSE = 1.08 m) than for FDM2.3 ($r^2 = 0.98$ and RMSE = 0.88 m). This is no surprise, however, as Kuipers Munneke et al. (2015) used the vertical density profiles of locations with $R_{MA} < 0.05$ to introduce a correction factor for the densification equations. For comparison purposes, we chose to not repeat this calibration procedure here, leading to a slight overestimation of density in the dry snow zone.

For locations with moderate melt (Figures 1C-F), both $r^2$ (0.87 to 0.92) and RMSE (2.81 m to 1.70 m) show a significant improvement from FDM2.3 to FDM2.3p2. This is mainly caused by the surface melt reduction in the RACMO2.3p2 forcing, resulting in less meltwater refreezing and therefore less dense firn columns. In Figure 1A, the open circles show the underes-timation of FAC in FDM2.3, which is much improved in FDM2.3p2 (closed circles). Another reason for denser firn columns in FDM2.3 is an artefact in the temperature-dependent part of the densification equation reported previously by Steger et al. (2017a). In this equation, the firn densification rate is overestimated when the vertically integrated temperature far exceeds the average surface temperature. In Greenland, this led to unrealistically high densification rates in the percolation zone and subse-quently too low FAC. In FDM2.3p2, this artefact was solved by replacing the average surface temperature in the densification equation with the temperature of the lowest model layer to account for the additional latent heat of refrozen water.

For the last category -locations with $R_{MA} > 0.5$-, both IMAU-FDM simulations underestimate observed FAC (Figure 1A). The simulated FAC of ∼0.5 m is typical for the model ablation zone at the end of winter, i.e. bare ice covered by a winter snow layer, while the observations suggest that firn of multiple years should be present with FAC varying between 1-4 m. This underestimation in FAC could be caused by remaining biases in atmospheric forcing or processes that are currently not



represented in IMAU-FDM (see below). Theory confirms that a firn layer should be present for $R_{MA}$ as large as $\sim$0.7 (Pfeffer et al., 1991).

Figures 2 and 3 confirm that the largest differences between FDM2.3 and FDM2.3p2 are found in the percolation zone of the GrIS. Along a transect in the percolation zone of the western GrIS (Harper et al., 2012), it is clear that the firn line (FL, defined

as $R_{MA} = 0.7$) is simulated further downslopw in FDM2.3p2 (Figure 2A-D). From observed FAC, the FL is located around 48.7 $^o$W, which is almost matched by FDM2.3p2 ($\sim$48.3 $^o$W), while FDM2.3 simulates the area where no firn is present up to $\sim$47.5 $^o$W (30 km further inland). Due to the reduction of surface melt in FDM2.3p2, a firn layer is formed at lower elevations.

The remaining discrepancy between the observations and FDM2.3p2, especially for $R_{MA} > 0.5$ (Figure 1A), is likely caused by how IMAU-FDM treats the vertical transport of liquid water. Currently, a 'tipping-bucket' method is used, assuming that

water can only run off if both cold content and pore space are unavailable. From observations however, it is found that through heterogeneous percolation (Humphrey et al., 2012) and/or impermeable ice lenses (Machguth et al., 2016), water can run off before all cold content or pore space is used.

Firn temperature is another useful metric to evaluate the performance of IMAU-FDM, especially in locations with substantial surface melt. The amount and depth of refreezing determines to a large extent how much heat is stored in the firn column, i.e.

how much $T_{10m}$ deviates from $T_s$. FDM2.3p2 shows much improved agreement with observed $T_{10m}$ (Figures 2E-F). For the eastern firn cores, realistic firn columns are simulated by both FDM2.3 and FDM2.3p2 with similar deep-firn temperatures as observed. Further west, FDM2.3 simulates lower temperatures than observed, indicating the absence of a firn layer that can store the heat released by refreezing. In FDM2.3p2, a band of higher firn temperatures (around -4 $^o$C) is simulated upslope of the FL, in good agreement with observed temperatures.

When the differences between FDM2.3 and FDM2.3p2 across the entire GrIS are considered (Figure 3), a clear pattern emerges. The largest differences in both FAC and $T_{10m}$ are located in the percolation zone of the GrIS and are dominated by the decrease in meltwater refreezing. This results in a FAC increase of 5-15 m and a downslope migration of the $T_{10m}$-band of high temperatures. In the higher elevation regions of the percolation zone, $T_{10m}$ dropped by 2-4 $^o$C due to the decrease in surface melt and subsequent refreezing and latent heat release, while in the lower percolation zone the presence of a simulated

firn layer in FDM2.3p2 results in much higher $T_{10m}$. The largest differences are found in southeast Greenland, where the influence of the previously mentioned temperature artefact in the densification equation is also significant as the firn is close to freezing in these firn-aquifer areas. Solving this issue resulted in lower densification rates and therefore thicker firn layers (i.e. high FAC) that are able to store the liquid water year-round as deep firn temperatures are at the freezing point (Figure 3E). The extent of the firn aquifer is therefore greatly improved in FDM2.3p2, compared to the results presented in Steger et al. (2017a).

In the ice sheet interior, the differences between FDM2.3 and FDM2.3p2 are a direct consequence of the atmospheric forcing: the increased accumulation results in faster densification and 2-3 m lower FAC, while the $T_{10m}$ increase is almost identical in magnitude and spatial pattern to the increase in $T_{2m}$ from RACMO2.3 to RACMO2.3p2 (not shown). The lowest regions of the GrIS show no differences in FAC, as it is an ablation area in both model simulations. For $T_{10m}$ however, FDM2.3p2 simulates 1-2$^o$C higher temperatures in the ablation zone, caused by a shorter presence of bare ice at the surface (i.e. less insulating

effect of a snow/firn layer). Over 1990–2009, FDM2.3p2 simulates 20 days yr$^{-1}$ (25%) less bare-ice exposure than FDM2.3.





Averaged over the entire GrIS (using only grid cells that are present in both FDM2.3 and FDM2.3p2), the $T_{10m}$ difference is +0.94$^o$C and the FAC difference is +1.13 m (8 %). The latter corresponds to a volume difference of roughly 2,000 km$^3$ and is equivalent to 11 years of meltwater storage at the 1960–1990 refreezing rate.

## 4  Conclusion

It is shown that the firn layer on the GrIS is highly sensitive to the forcing climate, mainly surface melt and accumulation. Improved atmospheric forcing (increased inland snowfall and decreased surface melt) from RACMO2, version 2.3p2, leads to significant improvements in simulated FAC and $T_{10m}$ in the percolation zone. In the interior dry snow zone and the ablation zone no large changes are found. The results suggest that the Greenland firn layer contains more pore space than previously thought, which has important implications for the liquid water retention capacity of the GrIS. A higher buffering capacity to

retain liquid water by either refreezing or storage is especially important if present-day firn conditions are used as starting point for future simulations, as it will delay and reduce the increase in runoff in a future warming climate. Data from the full simulation of Greenland firn density, temperature, and liquid water content at high temporal (10 days) and spatial resolution (11 km) are available for the period 1960–2016.

*Data availability.*   Modelled time series of firn air content and 10-m firn temperature are available on Pangaea, https://doi.pangaea.de/10.1594/PANGAEA.

All other IMAU-FDM output is available from the authors without conditions.

*Competing interests.*   The authors declare that there are no competing interests

*Acknowledgements.*   All authors acknowledge support of the Polar Program of the Netherlands Organisation for Scientific Research (NWO) and the Netherlands Earth System Science Centre (NESSC). Stefan Ligtenberg is supported by an NWO-ALW Veni grant, number 863.15.023.





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





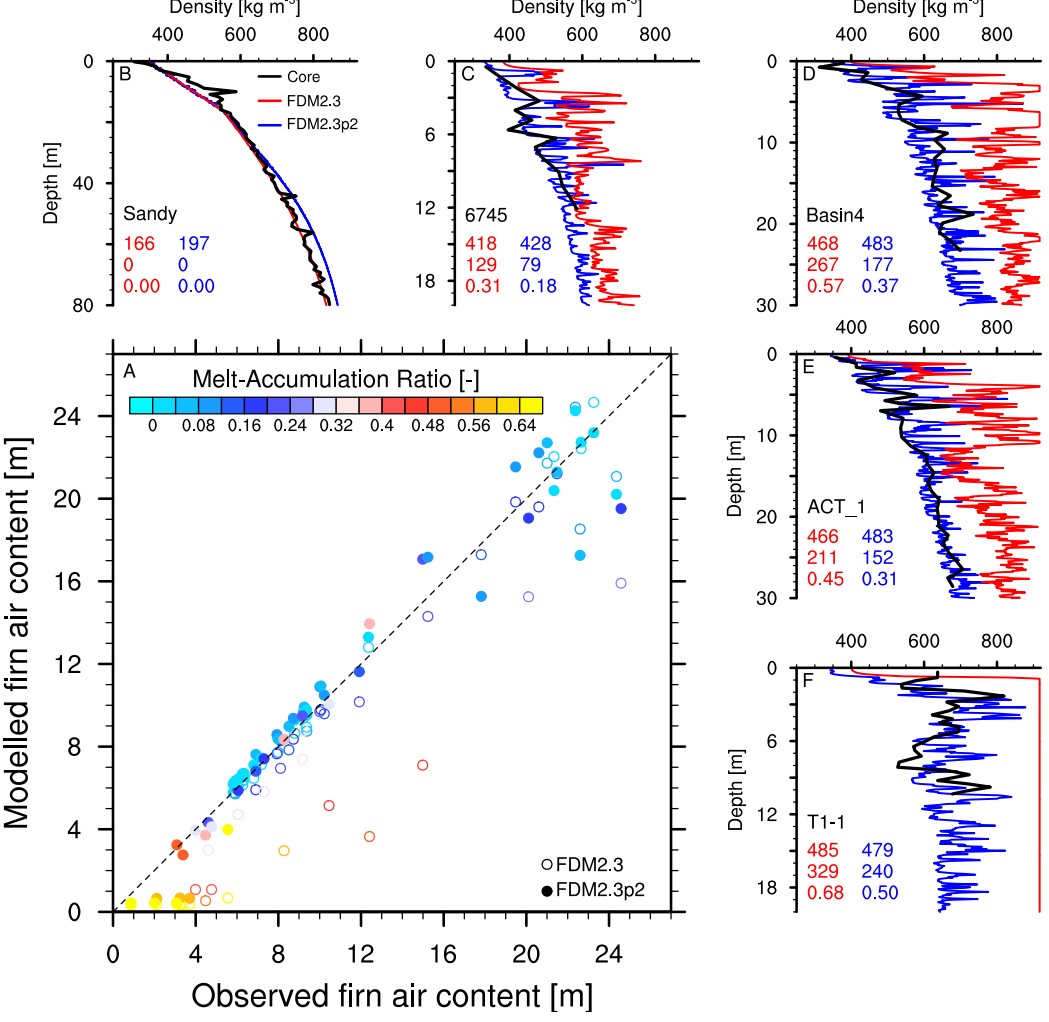

**Figure 1.** Evaluation of simulated firn density: (A) modelled vs. observed firn air content for FDM2.3 (open circles) and FDM2.3p2 (closed circles) at 62 firn core locations on the GrIS; (B-F) vertical firn density profiles of 5 selected cores (black), FDM2.3 simulation (red), and FDM2.3p2 simulation (blue). The colours in (A) represent the melt-accumulation ratio of the core location. In (B-F), the core name (black print) is provided, as well as the 1990–2009 average accumulation (in mm w.e. yr$^{-1}$), 1990–2009 average surface melt (in mm w.e. yr$^{-1}$), and the 1990–2009 melt-accumulation ratio (unitless) as simulated by RACMO2.3 (red print) and RACMO2.3p2 (blue print). Core names and locations can be found in Figure 2 of Kuipers Munneke et al. (2015).





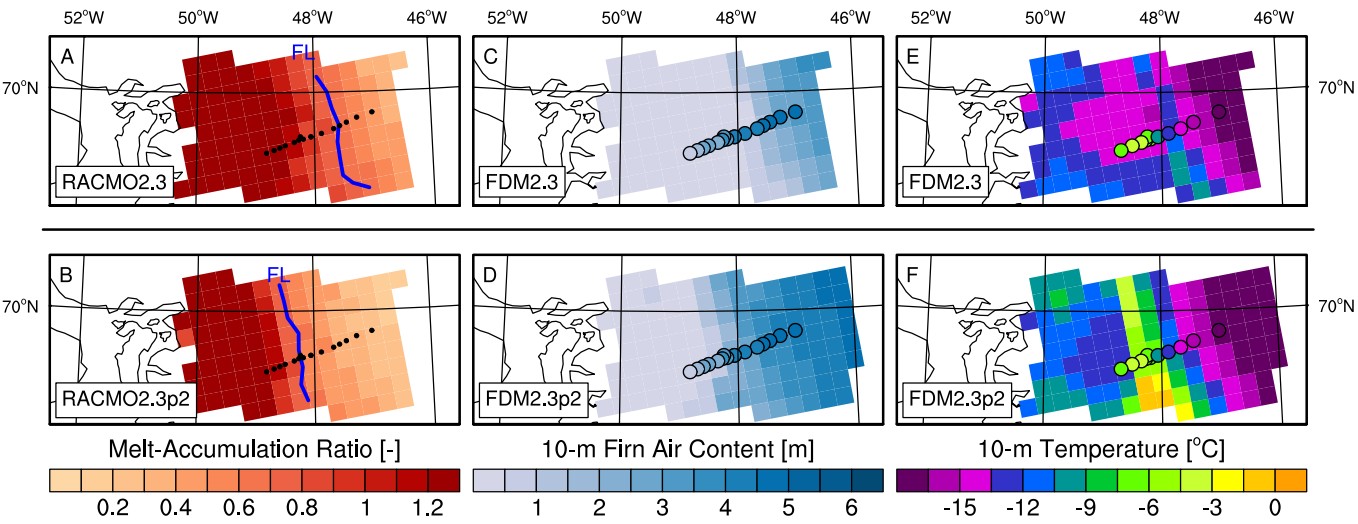

**Figure 2.** Evaluation of simulated firn air content and 10-m firn temperature with observations along a transect in the west Greenland (region indicated in Figure 3A) percolation zone: (A-B) 1990–2009 melt-accumulation ratio ($R_{MA}$) as simulated by RACMO2, (C-D) upper 10-m firn air content as simulated by IMAU-FDM (shaded grid cells) and from firn core observation (circles, Harper et al. (2012)); (E-F) average 10-m firn temperature as simulated by IMAU-FDM (shaded grid cells) and from thermistor string measurements (circles, Humphrey et al. (2012)). The figures in A-B represent RACMO2.3 and RACMO2.3p2, while C&E and D&F represent FDM2.3 and FDM2.3p2, respectively. Blue lines in A-B indicate the firn layer (FL), chosen to be equal to $R_{MA} = 0.7$. Firn core observations in C-D are from July 2007 or May 2008 and the simulated field is an average of these two dates. Both the simulated and observed firn temperatures in E-F are averages over 2007-2009.





**Figure 3.** Firn air content (FAC) and 10-m temperature ($T_{10m}$) as simulated by the IMAU-FDM: (A) FAC as simulated by FDM2.3; (B) FAC as simulated by FDM2.3p2; (C) the difference in FAC between FDM2.3 and FDM2.3p2; (D-F) similar to Figures A-C only for $T_{10m}$ instead of FAC. Box in A indicates the region used in Figure 2.