# Peer review of "Brief communication: Improved simulation of the present-day Greenland firn layer (1960-2016)"

_The Cryosphere, 2017_

## Referee Comment (RC1) · Anonymous Referee #1 · 6 Feb 2018

The study presents the results of the IMAU-FDM firn densification model forced with an updated set of boundary conditions, namely the RACMO2.3p2. As a result mainly increased snowfall inland and decreased surface melt in the input fields, FDM-simulated density profiles, subsurface temperatures and integrated firn air content are improved considerably compared to those driven by the previous version of RACMO2. While the study does not have groundbreaking results, it is nevertheless an important documentation of a widely used firn product across the disciplines. It also provides a useful illustration of the importance of weather forcing, and the potential perils of tuning a firn model to observed quantities if the problem lies in the weather forcing.

[Figure]

The manuscript is concise and well-written and the figures and analyses nicely support the conclusions. I suggest to accept with only minor revisions.

**MINOR POINTS ## P2L26-28: You discuss the downscaling to 1 km by Noël et al (2017) but do you use this in this paper? I cannot see that you do, and to avoid confusion, I suggest to leave this sentence out.**

P3L13-14+Fig 1: You discuss the three categories of the melt-accumulation ratio and Figure 1 has this quantity color-coded. But it is tricky to read off the colorbar. I suggest you choose a colorbar with three color-sets (eg. greens, blues and reds) that shifts exactly with the three categories.

P3L22-26: You list two reasons for improvement in the firn air content – reduced melt and fix of an artefact in the densification parameterization. You point to the former as the main reason, but how have you separated the two?

P4L5: downslope

P4L29: You mention that the extent of the firn aquifer is greatly improved, but you do not show or document this here, do you?

P4L34: You talk of higher temperatures in the ablation zone caused by shorter bare-ice duration and mention less insulating effect of a snow layer. I don't understand this – won't a shorter bare-ice duration (with an accompanying longer snow cover duration) lead to an increased insulating effect? Please review this sentence.

Fig 1 caption: Note that modeled profiles are taken at same time as the cores were drilled. Perhaps indicate on the profiles when this is.

Fig 2 caption: "firn layer (FL)" -> "firn line (FL)"

Fig 3 caption: "Difference between" can sometimes be a bit unclear. Please indicate exactly what is subtracted from what.

---

## Referee Comment (RC2) · C. M. Stevens (Referee) · 15 Feb 2018

Summary:

The paper presents an updated dataset of modeled firn air content and firn temperature for Greenland using the IMAU-FDM. The firn-densification model is the same as has been used for previous studies, but the model is forced using updated fields from the RACMO2.3p2 regional climate model. The paper demonstrates that using the updated forcing results in better model-data agreement in the zones that experience moderate summer melt.

[Figure]

General Comments:

Overall, this is a well-written and well-organized paper, and it should be of interest to many members of the ice-sheet mass balance scientific community. Firn-model simulations are important for understanding the mass balance of Greenland, especially as the area which experiences summer melt grows, and the IMAU-FDM simulations are used by many research groups. These updated model products are thus a valuable contribution.

The paper makes a convincing case that the new results are indeed improved, and I recommend this manuscript for publication with minor revisions.

General points to address:

- In the abstract, it may be useful to clarify that the improvement is a result of improved atmospheric forcing data, not improved model physics (line3).

- (general curiosity; does not necessarily need to be addressed in the paper): The RACMO data begin in 1958; why do your model simulations begin in 1960?

- Firn air content (FAC) is the metric of choice. A few things to consider: When you report FAC for a site (or the whole ice sheet, as in Figure 3), it is important to note to what depth you are modeling. For instance, some groups' firn model domains do not extend to the depth where density becomes 917. For example, if considering Summit, the FAC at ∼80 m depth is ∼22 m and at ∼200 m depth is ∼25 m.

- You are reporting the r2 and RMSE (page 3, line 17), but can you expand on how you are generating those statistics? Is it how well the 1-1 line in figure 1A fits the dots, and RMSE is the error there? Or, is the r2 and RMSE calculated for each model depth/density profile compared to the data? If it is the former, how is RMSE skewed by cores that were not drilled to the firn-ice transition (related to the point above), or do you only consider full-thickness cores? For instance, a FAC RMSE error of 1.08 m might be small if you are considering cores with full FAC of 20m, but quite large if it is

from a 10-m core with only 5 m of observed FAC. Would there be a way of normalizing the cores for this metric?

- Considering the comparison of modeled 10-m FAC and 10m firn temperatures to the Harper data. Can you provide a more quantitative description of the model-data mismatch for RACMO2.3 and 2.3p2 simulations? I can clearly see the difference in Figure 2 but some metric for the difference would be appreciated. Also, why does 2.3p2 still predict a very cold 10m temperature zone (blue/purple in Fig 2F) at the western edge of the data, where the data do not show that?

Specific/technical corrections:

Page 1, Line 12: continues: change to "will continue"

Page 1, Line 19: perhaps IMAU-FDM should be written out prior to the acronym being used.

Page 1, Line 19: change sentence from passive voice: Kuipers Munneke et al. (2015 simulated the temporal . . . firn layer using the IMAU-FDM.

Page 1, Line 25: you say "more accumulation inland and less surface melt" - less surface melt where? Also inland? Ice-sheet wide?

Page 2, Line 11: perhaps specify here that liquid water percolation is modeled using a bucket scheme (you mention it later, but may be appropriate here)

Page 2, Line 25: please define the area you mean by inland. Above a certain elevation? KM from the coast?

Page 4, Line 5: downslope misspelled.

Figure 1:

- I think that instead of referring the reader to another paper to find the site locations, you could include them (the 5 plotted here, at least) on one of the panels in figure 3.

-Do you have supplementary figures showing the improved/new modeled profiles for all 62 cores? I think it would be good if those were available somewhere.

- I know space is tight but having labels for the rows of numbers in the subpanels would be very useful to me. They could be as simple as b_dot, m_dot , and m_dot/b_dot.

- Since you have divided the firn into 3 regions (dry, moderate melt, high melt) it may be useful to choose a colormap that has 3 distinct zones, or to at least mark on the colorbar where the transitions between zones are.

- Since FAC is the metric you are looking at elsewhere, consider changing panels b-f to show FAC as a function of depth rather than density.

Figure 2:

- Can you show the location of the observed firn line in panels A and B for comparison to the modeled?

Figure 3:

- The color scale for the difference plots is a bit challenging because it is not linear; it does not clearly demonstrate your point that the biggest changes in FAC are in the moderate melt zone because the interior has some dark blue, but that is not nearly the magnitude of the red it turns out.

---

## Author Response (AR1)

Dear editor and reviewers,

We would like to thank the two reviewers for their positive and constructive comments, which improved the manuscript. Our response to the comments is written in *italic* and when text in the manuscript was changed/added it is provided in quotes. The updated Figures 1 and 3 are given at the end of the responses.

Kind regards,
Stefan Ligtenberg, on behalf of the authors

**Reviewer #1:**

**## MINOR POINTS ##**
P2L26-28: You discuss the downscaling to 1 km by Noël et al (2017) but do you use this in this paper? I cannot see that you do, and to avoid confusion, I suggest to leave this sentence out.

*Response: The 11-km ice-sheet integrated SMB is actually somewhat compromised in RACMO2.3p2, compared to RACMO2.3. This pertains to the representation of low-lying ablation zones only: these are underrepresented at 11-km resolution. Therefore, the 1-km downscaling technique was used and we find it valuable to leave this information in the manuscript. For clarity, we added the following sentence: "Here, the 11-km data was used as it is computationally not feasible to use the 1-km data."*

P3L13-14+Fig 1: You discuss the three categories of the melt-accumulation ratio and Figure 1 has this quantity color-coded. But it is tricky to read off the colorbar. I suggest you choose a colorbar with three color-sets (eg. greens, blues and reds) that shifts exactly with the three categories.

*Response: changed the colour bar.*

P3L22-26: You list two reasons for improvement in the firn air content – reduced melt and fix of an artefact in the densification parameterization. You point to the former as the main reason, but how have you separated the two?

*Response: No, we have not separated these two in a quantitative sense. Following the conclusion of Steger et al., 2017, sensitivity simulations were performed to investigate the influence of the artefact in the densification rate. It was found to only produce substantial differences in southeast Greenland, as mentioned in page 4, line 25-30. For the other regions of the ice sheet the difference between FDM2.3 and FDM2.3p2 are mainly related to differences in the forcing. We considered to remove the artifact statement from page 3, line 22-26, but decided to leave it in as we find it important to state that the FDM2.3p2 simulation does not include the artifact/bug reported by Steger et al. 2017.*

P4L5: downslope
*Response: Done*

P4L29: You mention that the extent of the firn aquifer is greatly improved, but you do not show or document this here, do you?

*Reply: Correct, we added "(not shown)" to the sentence. Within the brief communication format, it was not possible to include a figure showing the firn aquifer extent.*

P4L34: You talk of higher temperatures in the ablation zone caused by shorter bare-ice duration and mention less insulating effect of a snow layer. I don't understand this – won't a shorter bare-ice duration (with an accompanying longer snow cover duration) lead to an increased insulating effect? Please review this sentence.

*Response: Yes, you are correct. When surface melt is reduced, snow/firn remains present at the surface longer resulting in a shorter bare-ice duration. As snow/firn is present*

*at the surface longer, the insulation effect is longer. We replaced "less" with "increased".*

Fig 1 caption: Note that modeled profiles are taken at same time as the cores were drilled. Perhaps indicate on the profiles when this is.
> *Response: Added the year when the cores were drilled.*

Fig 2 caption: "firn layer (FL)" -> "firn line (FL)"
> *Response: Done*

Fig 3 caption: "Difference between" can sometimes be a bit unclear. Please indicate exactly what is subtracted from what.
> *Response: Done*

**Reviewer #2:**

**General points to address:**
In the abstract, it may be useful to clarify that the improvement is a result of improved atmospheric forcing data, not improved model physics (line3).
> *Response: removed "improved"*

(general curiosity; does not necessarily need to be addressed in the paper): The RACMO data begin in 1958; why do your model simulations begin in 1960?
> *Response: Yes that is correct, RACMO2 data begins in September 1958 similar to the forcing data of ERA-Interim. The choice for 1960 is twofold. First, the initial 1958 RACMO2 snowpack needs some time (months, year) to equilibrate with the simulated climate, making the 1958/59 near-surface climate by RACMO2 not the most reliable. Second, for the spin-up procedure of IMAU-FDM it feels more appropriate to use full years/decades. Therefore, we choose to use the period 1960-1979 as spin-up period.*

Firn air content (FAC) is the metric of choice. A few things to consider: When you report FAC for a site (or the whole ice sheet, as in Figure 3), it is important to note to what depth you are modeling. For instance, some groups' firn model domains do not extend to the depth where density becomes 917. For example, if considering Summit, the FAC at ~80 m depth is ~22 m and at ~200 m depth is ~25 m.
> *Response: The definition of FAC as given in section 2.3 of the manuscript correctly indicates how FAC is calculated in IMAU-FDM. IMAU-FDM simulates the firn density until a density where the ice density (917 kg m-3) is reached and FAC is the vertically integrated difference between firn density and ice density. We added: "In IMAU-FDM, all simulated firn layers extend to below the depth at which the ice density is reached, resulting in modelled FAC to represent the full firn column."*

You are reporting the r2 and RMSE (page 3, line 17), but can you expand on how you are generating those statistics? Is it how well the 1-1 line in figure 1A fits the dots, and RMSE is the error there? Or, is the r2 and RMSE calculated for each model depth/density profile compared to the data? If it is the former, how is RMSE skewed by cores that were not drilled to the firn-ice transition (related to the point above), or do you only consider full-thickness cores? For instance, a FAC RMSE error of 1.08 m might be small if you are considering cores with full FAC of 20m, but quite large if it is from a 10-m core with only 5 m of observed FAC. Would there be a way of normalizing the cores for this metric?
> *Response: It is the latter: the statistics are calculated for each model FAC compared to the observed FAC. Also, the statistics are calculated over all cores (full and partial). We revised the sentence to: "Overall, the agreement with observed FAC in the dry snow zone is slightly worse for FDM2.3p2 ($r^2 = 0.98$ and RMSE = 1.08 m) than for FDM2.3 ($r^2 = 0.98$*

*and RMSE = 0.88 m) for all cores combined". Since the statistics that are compared between FDM2.3 and FDM2.3p2 cover the same observed data we see no need to normalize the cores. In our opinion this would lead to less clear figures, while the statistics are currently only used to quantify the differences/improvements that are clearly visible in the figure.*

Considering the comparison of modeled 10-m FAC and 10m firn temperatures to the Harper data. Can you provide a more quantitative description of the model-data mismatch for RACMO2.3 and 2.3p2 simulations? I can clearly see the difference in Figure 2 but some metric for the difference would be appreciated. Also, why does 2.3p2 still predict a very cold 10m temperature zone (blue/purple in Fig 2F) at the western edge of the data, where the data do not show that?

> *Response: Thanks for the suggestion; we added RMSE and $r^2$ for both FAC and T10m in the text. "Quantitatively, FAC as simulated by FDM2.3p2 ($r2 = 0.71$ and RMSE = 1.64 o C) also shows much better agreement than in FDM2.3 ($r2 = 0.40$ and RMSE = 2.83 o C)" and "FDM2.3p2 ($r2 = 0.39$ and RMSE = 3.55 o C) shows much improved agreement over FDM2.3 ($r2 = 0.01$ and RMSE = 6.57 o C) for observed T10m (Figures 2E-F)".*

**Specific/technical corrections:**
Page 1, Line 12: continues: change to "will continue"
> *Response: changed.*

Page 1, Line 19: perhaps IMAU-FDM should be written out prior to the acronym being used.
> *Response: The written-out form of IMAU-FDM would be IMAU firn densification model. We feel "Institute for Marine and Atmospheric research Utrecht" is too long to add to this sentence.*

Page 1, Line 19: change sentence from passive voice: Kuipers Munneke et al. (2015) simulated the temporal … firn layer using the IMAU-FDM.
> *Response: changed.*

Page 1, Line 25: you say "more accumulation inland and less surface melt" – less surface melt where? Also inland? Ice-sheet wide?
> *Response: added "ice-sheet wide".*

Page 2, Line 11: perhaps specify here that liquid water percolation is modeled using a bucket scheme (you mention it later, but may be appropriate here)
> *Response: added this.*

Page 2, Line 25: please define the area you mean by inland. Above a certain elevation? KM from the coast?
> *Response: added "(i.e. accumulation area)"*

Page 4, Line 5: downslope misspelled.
> *Response: changed.*

**Figure 1:**
- I think that instead of referring the reader to another paper to find the site locations, you could include them (the 5 plotted here, at least) on one of the panels in figure 3.
> *Response: Added the 5 locations and names in Fig 3A.*

-Do you have supplementary figures showing the improved/new modeled profiles for all 62 cores? I think it would be good if those were available somewhere.
> *Response: Since this manuscript will be published as a brief communication, we decided to not include such a figure in the manuscript or as supp. figure.*

- I know space is tight but having labels for the rows of numbers in the subpanels would be very useful to me. They could be as simple as b_dot, m_dot , and m_dot/b_dot.

   *Response: Added "Acc", "Me", and $R_{MA}$ for clarity.*

- Since you have divided the firn into 3 regions (dry, moderate melt, high melt) it may be useful to choose a colormap that has 3 distinct zones, or to at least mark on the colorbar where the transitions between zones are.

   *Response: changed this.*

- Since FAC is the metric you are looking at elsewhere, consider changing panels b-f to show FAC as a function of depth rather than density.

   *Response: Here, FAC is only used as a metric to describe the entire firn column with one value. Also a figure showing a vertical profile of FAC(z) is probably more difficult to interpret for readers. Therefore, we decided to keep these panels as is.*

**Figure 2:**
- Can you show the location of the observed firn line in panels A and B for comparison to the modeled?

   *Response: we are not aware that a dataset with the observed firn line is available. To determine a firn line, one would need a transect of SMB stakes (e.g. K-transect) or it could be mapped by satellite (e.g. MODIS). In this case, the observed FAC (Figure 2C-D) gives some indication on where the observed firn line is located. It is likely located slightly downstream of the FDM2.3p2-simulated firn line.*

**Figure 3:**
- The color scale for the difference plots is a bit challenging because it is not linear; it does not clearly demonstrate your point that the biggest changes in FAC are in the moderate melt zone because the interior has some dark blue, but that is not nearly the magnitude of the red it turns out.

   *Response: changed the color scale of Fig 3C. Now, the red in the southeast is clearly darker than the blue in the interior.*

[Figure]

Revised Figure 1.

[Figure]

Revised Figure 3

[revised manuscript text omitted]

Figures 2 and 3 confirm that the largest differences between FDM2.3 and FDM2.3p2 are found in the percolation zone of the GrIS. Along a transect in the percolation zone of the western GrIS (Harper et al., 2012), it is clear that the firn line (FL, defined as $R_{MA} = 0.7$) is simulated further downslope in FDM2.3p2 (Figure 2A-D). From observed FAC, the FL is located around 48.7 °W, which is almost matched by FDM2.3p2 (∼48.3 °W), while FDM2.3 simulates the area where no firn is present up to ∼47.5 °W (30 km further inland). Due to the reduction of surface melt in FDM2.3p2, a firn layer is formed at lower elevations. Quantitatively, FAC as simulated by FDM2.3p2 ($r^2 = 0.71$ and RMSE = 1.64 °C) also shows much better agreement than in FDM2.3 ($r^2 = 0.40$ and RMSE = 2.83 °C).

[revised manuscript text omitted]
 , where green, blue and red colours indicate the three categories as specified in Section 3 . In (B-F), the core name and date (black print) is provided, as well as the 1990–2009 average accumulation ("Acc" in mm w.e. yr$^{-1}$), 1990–2009 average surface melt ("Me" in mm w.e. yr$^{-1}$), and the 1990–2009 melt-accumulation ratio ($R_{MA}$ , unitless) as simulated by RACMO2.3 (red print) and RACMO2.3p2 (blue print). Core names and locations can be found in Figure 3A.

[Figure]

**Figure 2.** Evaluation of simulated firn air content and 10-m firn temperature with observations along a transect in the west Greenland (region indicated in Figure 3B) percolation zone: (A-B) 1990–2009 melt-accumulation ratio ($R_{MA}$) as simulated by RACMO2, (C-D) upper 10-m firn air content as simulated by IMAU-FDM (shaded grid cells) and from firn core observation (circles, Harper et al. (2012)); (E-F) average 10-m firn temperature as simulated by IMAU-FDM (shaded grid cells) and from thermistor string measurements (circles, Humphrey et al. (2012)). The figures in A-B represent RACMO2.3 and RACMO2.3p2, while C&E and D&F represent FDM2.3 and FDM2.3p2, respectively. Blue lines in A-B indicate the firn line (FL) , chosen to be equal to $R_{MA} = 0.7$. Firn core observations in C-D are from July 2007 or May 2008 and the simulated field is an average of these two dates. Both the simulated and observed firn temperatures in E-F are averages over 2007-2009.

[Figure]

**Figure 3.** Firn air content (FAC) and 10-m temperature ($T_{10m}$) as simulated by the IMAU-FDM: (A) FAC as simulated by FDM2.3; (B) FAC as simulated by FDM2.3p2; (C) the difference in FAC  (FDM2.3p2 minus FDM2.3) ; (D-F) similar to Figures A-C only for $T_{10m}$ instead of FAC.  Locations in A indicate the cores used in Figure 1B-F, while box in B indicates the region used in Figure 2.